# Mapping Fire Severity in Southwest China Using the Combination of Sentinel 2 and GF Series Satellite Images

**DOI:** 10.3390/s23052492

**Published:** 2023-02-23

**Authors:** Xiyu Zhang, Jianrong Fan, Jun Zhou, Linhua Gui, Yongqing Bi

**Affiliations:** 1Institute of Mountain Hazards and Environment, Chinese Academy of Sciences, Chengdu 610041, China; 2Sichuan Forestry and Grassland Inventory and Planning Institute, Chengdu 610081, China; 3Sichuan Chuanjian Geotechnical Survey Design Institute Co., Ltd., Chengdu 610000, China

**Keywords:** fire severity, high-resolution satellite images, GF series satellite, machine learning, random forest

## Abstract

Fire severity mapping can capture heterogeneous fire severity patterns over large spatial extents. Although numerous remote sensing approaches have been established, regional-scale fire severity mapping at fine spatial scales (<5 m) from high-resolution satellite images is challenging. The fire severity of a vast forest fire that occurred in Southwest China was mapped at 2 m spatial resolution by random forest models using Sentinel 2 and GF series remote sensing images. This study demonstrated that using the combination of Sentinel 2 and GF series satellite images showed some improvement (from 85% to 91%) in global classification accuracy compared to using only Sentinel 2 images. The classification accuracy of unburnt, moderate, and high severity classes was significantly higher (>85%) than the accuracy of low severity classes in both cases. Adding high-resolution GF series images to the training dataset reduced the probability of low severity being under-predicted and improved the accuracy of the low severity class from 54.55% to 72.73%. RdNBR was the most important feature, and the red edge bands of Sentinel 2 images had relatively high importance. Additional studies are needed to explore the sensitivity of different spatial scales satellite images for mapping fire severity at fine spatial scales across various ecosystems.

## 1. Introduction

Wildfires are a problematic and recurrent issue in Southwest China, which has attracted considerable attention from the government and researchers [1]. Wildfires cause severe loss of life and property and seriously affect the stability of forest ecosystems [2,3,4,5,6,7,8,9]. Discrimination of fire severity is the foundation of fire behavior analysis, ecological risk assessment, and rehabilitation planning [10,11].

Previous work has shown that machine learning classifiers are well suited to regional-scale fire severity mapping with remotely sensed images such as Landsat images, Sentinel 2 images, and unmanned aerial vehicle images [11,12,13,14,15,16]. Among numerous remote sensing indices derived for fire severity mapping, NBR-based indexes incorporating shortwave infrared (SWIR) bands, such as NBR [17], dNBR [18], and RdNBR [19,20], have been shown to map the spatial variation in fire severity. A combination of multiple types of information can be used by machine learning methods to obtain more accurate results than grading a single remote sensing index. In addition to the NBR-based indices, other remote sensing indices related to vegetation and soil, such as NDVI-based indices and BSI-based indices, can also be used [11]. Furthermore, the fraction of vegetation cover and evapotranspiration information can also be used as inputs for machine learning models [21].

Fire severity mapping using high-resolution satellite imagery can capture heterogeneous fire severity patterns and characterize the fire-driving ecological processes changes at fine spatial scales over large spatial extents [13,22]. Although numerous studies have been applied to fire severity mapping at 10 m (Sentinel 2 images based) and 30 m (Landsat series images based) spatial resolutions, regional-scale fire severity has still been seldom mapped at fine spatial scales (<5 m) from high-resolution satellite images. This is because the commonly used high-resolution satellite images do not have the shortwave infrared (SWIR) bands and thus cannot calculate NBR-based indices. Furthermore, the authors of [11] demonstrated that high canopy cover and topographic complexity were associated with a higher rate of under-prediction, especially in lower severity classes. High-resolution satellite images can view more information about the burnt understory vegetation through forest gaps, potentially improving the accuracy of lower severity classes mapping. Therefore, exploring remote sensing approaches that can map fire severity at fine spatial scales in fire-prone ecosystems is significant and challenging.

The recent increase in the availability of high-resolution satellite imagery provides an opportunity to assess fire severity at fine spatial scales. In 2010, China launched a “high-resolution earth observation systems” program, which includes seven different types of high-resolution earth observation satellites [23]. The first satellite, GF-1, was successfully launched on 26 April 2013 and can provide multi-resolution (2 m, 8 m, and 16 m) data with a 4-day revisit cycle. The GF-6 satellite, successfully launched in 2018, has the exact same resolution and revisit cycle as the GF-1 satellite. A 2-day revisit cycle can be achieved through the operation in conjunction with the on-orbit GF-1 satellite [24]. The suitable spatial resolution and revisit cycle of GF series satellites provide feasibility for fire monitoring.

Some studies have already used GF images for fire monitoring, such as fire line extraction [25]. However, studies of fire severity mapping using GF images are still lacking due to the lack of SWIR bands for computing NBR-based indices. This problem may be solved by combining GF series satellite and Sentinel 2 satellite images. The Sentinel 2 is a constellation of two satellites, with the first one launched in June 2015 and the second one launched in March 2017. The multispectral instrument onboard the Sentinel 2 constellation captures images across 13 spectral bands at 10 m and 20 m resolution with a 5-day revisit cycle at mid-latitudes. Previous studies showed that Sentinel 2 generally performed as well as or better than Landsat for fire severity mapping [26,27]. Due to the finer spatial resolution and richer spectrum, Sentinel 2 imagery was selected instead of Landsat series images. Furthermore, Sentinel 2 imagery was selected because its red-edge bands exhibited a high correlation with fire severity [28].

This study aimed to use high-resolution GF series satellite and Sentinel 2 satellite images to map fire severity at 2 m spatial resolution. The panchromatic band at 2 m resolution and the visible and near infra-red (NIR) bands at 8 m resolution of GF series images was selected to cooperate with the Sentinel 2 images. We evaluate the influence of GF series images on fire severity mapping and analyze the fire severity spatial distribution of a fire-prone artificially planted forest ecosystem in a mountainous region of Southwest China.

## 2. Study Area

The Lushan Mountain region in Xichang City, Liangshan Yi Autonomous Prefecture (Sichuan Province, Southwest China), was selected as the study area, as shown in Figure 1. On 30 March 2020, a vast forest fire occurred in the village of Jingjiu in Xichang City. The fire lasted for three days, causing 19 deaths and 3 injuries. The burned area exceeded 30 km^2^, completely covering the Lushan Mountain scenic area. The densely populated residential area of Xichang City is adjacent to the northern part of the study area. Xichang City has a subtropical monsoon climate, with rainy summers and dry winters. The study area is a typical artificially planted forest area in mountainous southwest China. The dominant tree species are fire-prone *Pinus Yunnanensis* and *Eucalyptus*, which were planted by aerial seeding from the 1950s to the 1960s. *Pinus Yunnanensis* is rich in flammable turpentine, and the understory herbs are abundant. Once a fire starts, a major forest fire can be formed. According to the available data, at least four large-scale fires have occurred in the artificially planted forest area of Xichang City since 2010, on 4 March 2010, 27 January 2012, 18 March 2014, and 30 March 2020.

## 3. Methodology

The methodology comprises four steps: data acquisition and pre-processing, machine learning modeling, accuracy assessment, and distribution analysis (Figure 2).

### 3.1. Data Acquisition and Pre-Processing

Multiple satellite images and ancillary data were used in the present work (Table 1). In this section, we first introduce the satellite images, followed by the additional data, including topographical data and vegetation data.

#### 3.1.1. Satellite Images

This study selected GF series and Sentinel 2 satellite images as close as possible to the fire’s start date (30 March 2020) and end date (1 April 2020). GF-1 and GF-6 images were provided by the High-Resolution Earth Observation System Sichuan Data and Application Center (https://www.cresda.com/ accessed on 4 May 2020). The Spatial resolution of panchromatic bands of GF-1 and GF-6 is 2 m. Blue, green, red, and near-infrared (NIR) bands of GF-1 and GF-6 were pan-sharpened from 8 m to 2 m resolution by the Gram–Schmidt method [29] after atmospheric correction and ortho-rectification. The differences in corresponding bands between sensors of GF-1 and GF-6, as shown in Table 2, were eliminated based on their spectral response functions. Level 1C products of Sentinel 2 were downloaded from the Copernicus Hub (https://scihub.copernicus.eu/ accessed on 16 May 2020). Red-edge and shortwave infrared (SWIR) bands of Sentinel 2 images were resampled from 20 m to 10 m resolution using the nearest neighbor resampling algorithm after atmospheric correction and BRDF normalization [30,31]. Bandpass adjustment was performed to minimize the differences between the sensors of Sentinel 2A and Sentinel 2B, as shown in Table 3. The high-resolution GF image was used as the reference for executing the image registration process. This ensured accurate alignment of the sentinel images to a consistent standard.

#### 3.1.2. Topographical Data and Vegetation Data

A digital elevation model (DEM) with a 5 m resolution was used in the ortho-rectification [32] of both GF series and Sentinel 2 images. The DEM was obtained through an aerial photogrammetry technique at a spatial resolution of 5 m using the linear interpolation algorithm. These topographical data are the most recent and highest-resolution terrain data available. Plot-based vegetation maps for 2019 and 2020 were produced by artificial interpretation of high-resolution images.

#### 3.1.3. Training and Validation Dataset

The training and validation dataset was obtained through a field investigation from 28 July 2020 to 30 July 2020. The field survey recorded GPS coordinates, vegetation types, and forest canopy and understory coverage to classify the fire severity of the sample points on site using the standard presented in Table 4. A total of 500 samples were collected, mostly distributed in unburnt and high severity classes. The low and moderate severity classes were naturally rare. For each sample, corresponding pixel values were extracted for each band reflectance of images, spectral indexes, and additional data to create a training and validation dataset used as input into the RF models. A subset of 80% of the samples was extracted randomly to train the two models, and the remaining 20% was used to validate the two models. The validation samples were independent of the training samples to avoid optimistic bias [33]. The spatial distribution of training and validation samples is shown in Figure 1. The samples were randomly and almost evenly distributed in the study area. The minimum spatial distance between any two samples was greater than 30 m, ensuring a single Sentinel 2 pixel was not sampled twice.

### 3.2. Machine Learning Modeling

GF series and Sentinel 2 images were selected to map the fire severity of the study area at 2 m spatial resolution. In order to validate the effects of high-resolution images on the classification accuracy of fire severity, a comparative experiment using the random forest (RF) method [34] was carried out based on the Python sci-kit learn library (github.com/scikit-learn/scikit-learn accessed on 20 May 2020). One RF model (the S2-GF model) used GF series images, Sentinel 2 images, topographical data, and vegetation data as training and validation data. The other model (the S2 model) used Sentinel 2 images, topographical data, and vegetation data but did not use GF series images. The two models used the same samples to train and validate the classification models.

The features used in the two models to identify fire severity comprise reflectance, spectral indexes, and additional data, as shown in Table 5. The formula of spectral indexes of pre-fire and post-fire Sentinel 2 images, including NBR-based, NDVI-based, and BSI-based indexes, is shown in Table 6. The NDVI-based indices can reflect the vegetation change after the fire, and the BSI-based indices reflect the bare soil information after the fire. Due to the lack of SWIR bands, only NDVI-based spectral indexes were calculated using pre-fire and post-fire GF images. The spatial resolutions of the Sentinel 2 images, GF images, and DEM after pre-processing were 10 m, 2 m, and 5 m, respectively. And the vegetation type was vector data based on a forest compartments survey. In order to align the features with different spatial resolutions, all data are resampled to 1 m resolution using the nearest neighbor resampling algorithm.

### 3.3. Classification Accuracy Assessment

The confusion matrix and the Kappa statistic [37] were used to calculate the classification accuracy for each class and compare the performance between the two models. The feature importance of each variable on random forest classification models was ranked to assess feature importance on classification accuracy.

### 3.4. Fire Severity Classes Distribution Analysis

After mapping the fire severity of the study area, all pixels in the fire severity map predicted by the RF model with higher accuracy were used to analyze the relationship between fire severity classes and topography. The dominant tree species are *Pinus Yunnanensis* and *Eucalyptus*, which belong to the evergreen needle leaf forest and sclerophyllous evergreen broadleaf forest, respectively. The area of each fire severity class predicted by the RF model was counted by vegetation type.

## 4. Results

### 4.1. Model Results and Accuracy Comparison

The optimal parameters of the models were determined by a cross-validated grid search over a parameter grid, which exhaustively considered all parameter combinations for given values. For the estimator’s number and max depth, lists of (50, 100, 200, 300, 500, 800, and 1000) and (3, 5, 8, 10, 15, and 20) were set, respectively in this study. The parameters were optimized to be 100 for the estimator’s number and 15 for the max depth in both models. The fire severity maps predicted by the RF models are shown in Figure 3. There were no significant differences in the spatial pattern of fire severity between the results of the two models. However, the S2-GF model predicted more accurately in the transitional areas where the fire severity changes from low to high (Figure 3). To prove that the constructed sample patterns are efficient for the random forest classification, evaluation metrics such as precision, recall, F1 score, confusion matrix, and Kappa statistic were used (Table 7). The confusion matrixes showed that the RF model with the higher Kappa score and higher overall accuracy was given by the S2-GF model (Table 7). The overall accuracy of the S2-GF model was 91%, which was 6% higher than the S2 model. Moreover, the S2-GF model outperformed the S2 model in every class of fire severity, especially in the low severity class (Table 7). There were five samples under-predicted, 91 samples correctly predicted, and 4 points over-predicted in the S2-GF model. In comparison, 6 samples were under-predicted, 85 samples were correctly predicted, and 9 points were over-predicted in the S2 model. The proportion of validation data across fire severity classes in each prediction type is shown in Figure 4. Adding GF data to the training dataset reduced the probability of low fire severity being under-predicted as the unburnt class from 50% to 40%. The relative decrease in the feature importance for the two models (trained with and without GF data) ranked the features as shown in Figure 5. RdNBR was the most important feature in both models. Nine variables in the top ten variables in the two models were consistent, but their importance rankings were different. The post-fire NDVI of GF images was the only feature in the top ten associated with GF data.

### 4.2. Relationship between Fire Severity Classes and Topography

All pixels in the fire severity map predicted by the S2-GF model were used to analyze the relationship between fire severity classes and topography. The median altitude of each severity class increased with the severity level, as shown in Figure 6. Figure 7 indicated that fire severity presented different patterns in different ranges of aspect and slope. Due to the fire spreading from the southwest to the northeast of the study area, a part of the southern part of the study area was unburnt (Figure 3). The low and moderate severity part was mainly located on higher slopes in the eastern part of the study area, where it was close to the lake and relatively humid.

### 4.3. Relationship between Fire Severity Classes and Vegetation Type

The study area is a typical artificially planted forest ecosystem in Southwest China, as indicated by manual interpretation results of GF images with an accuracy of 98.68%. Forest covers 89.78% of the total land in the study area, while other land use types such as cultivated land, shrubland, grassland, artificial surfaces, and water bodies occupy 4.70%, 1.68%, 2.57%, 1.13%, and 0.14%, respectively. Figure 8 shows the distribution of land use in the study area. The dominant tree species are *Pinus Yunnanensis* and *Eucalyptus*, accounting for 60.90% and 15.94% of the total area of the study area, respectively. *Pinus Yunnanensis* is an evergreen needle-leaf forest, and *Eucalyptus* is a sclerophyllous evergreen broadleaf forest. Other tree species found in the study area, such as *Cyclobalanopsis*, aspen, and oak, primarily comprise evergreen broadleaf forests. The areas with higher altitudes are dominated by *Pinus Yunnanensis*, while *Eucalyptus* and the other vegetation were mainly distributed in the west of the study area with lower altitudes. *Pinus Yunnanensis* and *Eucalyptus* were more severely affected by the fire, with only 14.34% of *Pinus Yunnanensis* unburnt, compared to 30.06% of *Eucalyptus* and 49.17% of the remaining area (Table 8).

## 5. Discussion

### 5.1. Error Analysis

The results demonstrated that fire severity could be mapped with high accuracy (an overall accuracy of 85% by the S2 model and 91% by the S2-GF model) by random forest classification. The fire severity maps predicted by the two models had comparable accuracy to other studies [11,13,15,16]. No matter which model, the classification accuracy of unburnt, moderate, and high severity classes was significantly higher (>85%) than the accuracy of low severity classes (50–75%) (Table 7). This finding agrees with previous work [38] and may have been related to the limitations of the training and validation dataset. The accuracy pattern of the S2 model, as shown in Figure 4, was similar to the findings of previous work [11]. Of the under-predicted samples, the lower severity class represented the highest proportion. Of the over-predicted samples, the moderate severity class accounted for the highest proportion. Adding GF data to the training dataset reduced the probability of low fire severity being under-predicted as the unburnt class from 50% to 40%. It improved the accuracy of low severity classes from 54.55% to 72.73%. The post-fire vegetation situation was directly related to forest fire severity, because fire severity was defined as the loss of organic matter. Due to the higher resolution, there were fewer mixed pixels in the GF image than in the Sentinel 2 image, especially in the transitional areas where the fire severity changes from low to high (Figure 9). The post-fire NDVI of GF images provided high resolution vegetation information, which can significantly improve the accuracy of fire severity classes in areas with highly variable fire severity classes. Since the classification accuracy of the other three classes is relatively high, this improvement is more evident in the low severity class. In addition, the accuracy evaluation used a ground truth of the fire severity of 500 field-recorded samples. In regions with surveillance equipment, image annotation techniques can construct a large sample library from photos and videos to improve the accuracy of forest fire monitoring [39,40].

### 5.2. Feature Importance Comparison

The RdNBR of Sentinel 2 (S2) images was the most important feature in both models, which was consistent with other studies [11,14]. Although all of the reflectance bands were treated as features, only the post-fire NIR and red edge bands of S2 images were ranked in the top ten in both models. The closer the red edge band was to the NIR band, the higher its importance. The previous study demonstrated the superiority of red edge spectral indices over conventional spectral indices of S2 images [28]. Future improvements of features such as building NBR-based or NDVI-based indexes using red edge bands instead of NIR bands or SWIR bands may improve the models’ accuracy. In the S2-GF model, the importance of all of the reflectance bands of GF images failed to make it into the top ten. In spectral indexes features, RdNBR, post-fire NBR, dNDVI, dNBR, and post-fire NDVI of S2 images ranked the top ten in both models with different rankings. The importance of the post-fire NDVI of GF images was second only to the RdNBR of S2 images in the S2-GF model. In contrast, the post-fire BSI of S2 image features ranked in the top ten in the S2 model. According to previous studies, the NBR-based indices were the most popular in fire severity mapping [19,20,41]. The proportion change of vegetation and bare soil is directly related to fire severity. Therefore, NDVI- and BSI-based indices also had relatively high importance. The post-fire NDVI of GF images provided vegetation information with higher resolution, which was important for the model prediction in the transitional areas of fire severity. A high-resolution satellite in the future that has SWIR bands may further improve classification accuracy in areas with highly variable fire severity classes.

### 5.3. Fire Severity Distribution

Although the importance of topography and vegetation features did not rank in the top ten in both models (Figure 5), the spatial distribution of fire severity classes is related to elevation, slope, aspect, and vegetation type. As shown in Figure 6, the median altitude of each severity class increased with the severity level. This may relate to the spatial distribution of fire-prone *Pinus yunnanensis* and *Eucalyptus* in the study area. These fire-prone trees are mainly distributed in higher altitudes, while shrubland and cultivated land is distributed in low-altitude areas, such as valleys. The proportion of non-vegetative objects in the study area is small, accounting for just 1.27%. In areas where non-vegetative objects are more prevalent, it is recommended to include these objects, such as artificial surfaces, water bodies, and bare land, in the training samples for the fire severity mapping model. This will reduce the potential impact of these objects on the accuracy of the predicted fire severity maps. It is important to consider both vegetative and non-vegetative components in the study area and make necessary adjustments to the model to ensure optimal performance. Furthermore, the recovery speed of different vegetation types in the study area was different. According to the field survey in July 2020, the burnt grass had re-sprouted three months after the fire (Figure 1), while the burnt *Pinus yunnanensis* with high severity class was difficult to recover. In Southwest China, there is a large area dominated by artificially planted pine trees. The fire on March 30 2020 has attracted the attention of the academic community [25,42]. This study can provide an accurate fire severity map of this area at 2 m spatial resolution and an essential reference for fire severity mapping in similar regions.

## 6. Conclusions

In conclusion, this study presents an approach to map fire severity in Southwest China at 2 m spatial resolution using random forest models and satellite images from Sentinel 2 and GF series. The results demonstrated that (1) the S2-GF model, which used the combination of Sentinel 2 and GF series satellite images, had an overall accuracy of 91%, while the S2 model, which used Sentinel 2 images without GF images, had an overall accuracy of 85%. The S2-GF model showed some improvement (6%) in global classification accuracy relative to the S2 model. The increase in accuracy is more pronounced for the low severity class (from 54.55% to 72.73%) than for other classes. (2) RdNBR was the most important feature, and the red edge bands of Sentinel 2 images had relatively high importance in both models. The importance of the post-fire NDVI of GF images is second in the S2-GF model. (3) The fire severity in the mountainous region showed a gradient distribution with the altitude change. The median altitude of each severity class increased with the severity level. (4) Fire-prone tree species were more severely affected than other vegetation.

The results of this study highlight the potential of high-resolution satellite images for mapping fire severity at fine spatial scales. This study presents a machine learning approach that combines the strengths of multiple satellite images, providing a reference for future research. Looking ahead, the future improvement of the satellite sensors, especially the high-resolution sensors with SWIR bands, can further improve fire severity mapping accuracy. Continuing research to assess the sensitivity of different spatial scales of satellite images for mapping fire severity across various ecosystems is essential.

## Figures and Tables

**Figure 1 sensors-23-02492-f001:**
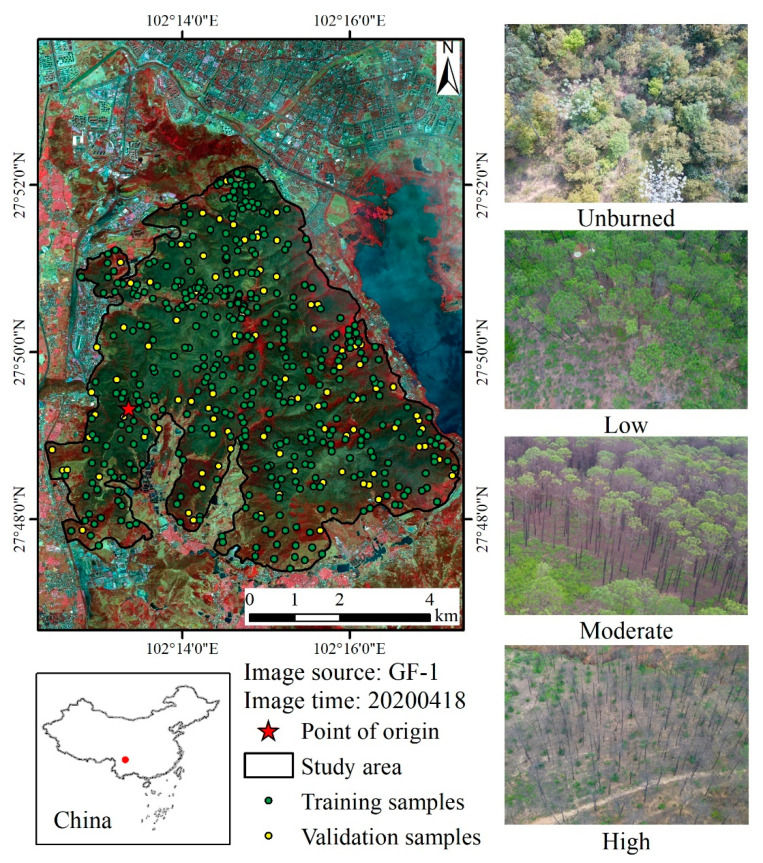
The location of the study area with field survey photos of different fire severity levels. The red dot indicates the location of the study area in China. The densely populated residential area of Xichang City, Liangshan Yi Autonomous Prefecture (Sichuan Province, Southwest China), is adjacent to the northern part of the study area. Due to the government’s restrictions on entering the mountain after the fire, the field photos of different fire severity classes were taken in July. The burnt understory grass had re-sprouted three months after the fire in July. Burn marks can be seen on the ground and tree trunks can be seen in the low severity field photo. Most of the canopy (>50%) was scorched in the moderate severity field photo. Full canopy biomass was consumed in the high severity field photo.

**Figure 2 sensors-23-02492-f002:**
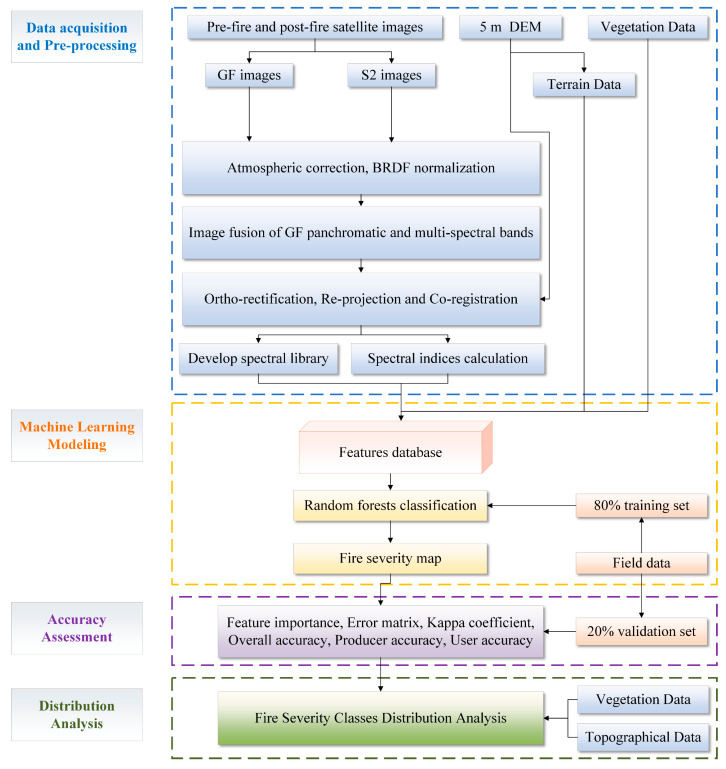
Flowchart of the methodology.

**Figure 3 sensors-23-02492-f003:**
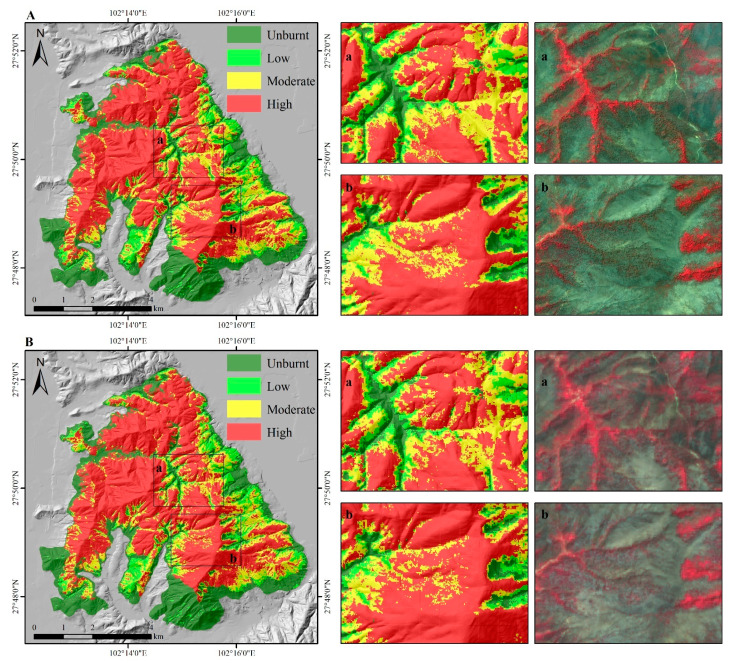
Fire severity maps. (**A**) Result of the S2-GF model and the GF-1 image on 18 April 2020; (**B**) result of the S2 model trained without GF data and the Sentinel 2 image on 4 April 2020. There were no significant differences in the spatial pattern of fire severity between the results of the two models. However, the S2-GF model trained with the combination of the Sentinel 2 data and GF data predicted more accurately, especially in the transitional areas where the fire severity changes from low to high (region a and region b in the figure).

**Figure 4 sensors-23-02492-f004:**
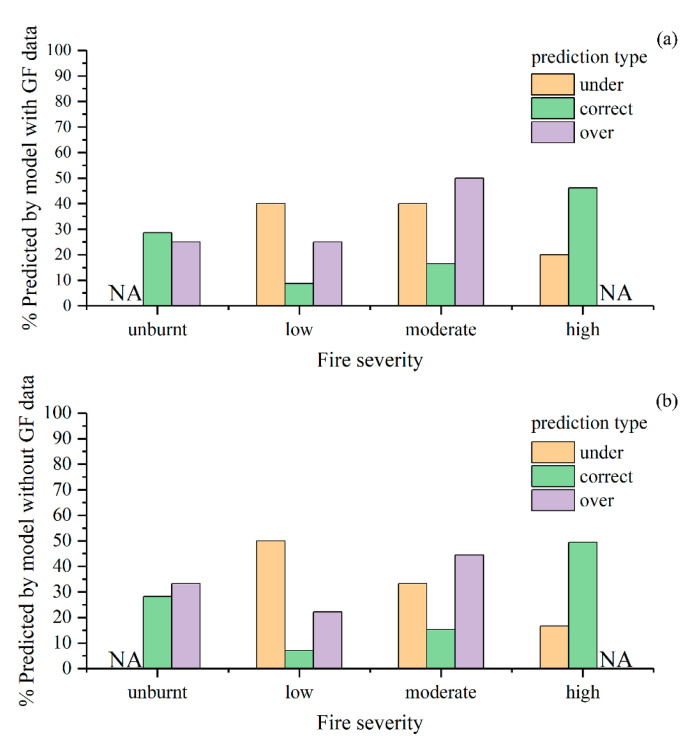
The proportion of validation data across fire severity classes in each prediction type of the S2-GF model (**a**) and the S2 model (**b**); under-predicted (the model predicted a lower severity class compared to the validation data), correctly predicted (the model predicted the same severity class compared to the validation data), and over-predicted (the model predicted a higher severity class compared to the validation data). Unburnt could not be under-predicted and extreme severity could not be over-predicted (i.e., NA).

**Figure 5 sensors-23-02492-f005:**
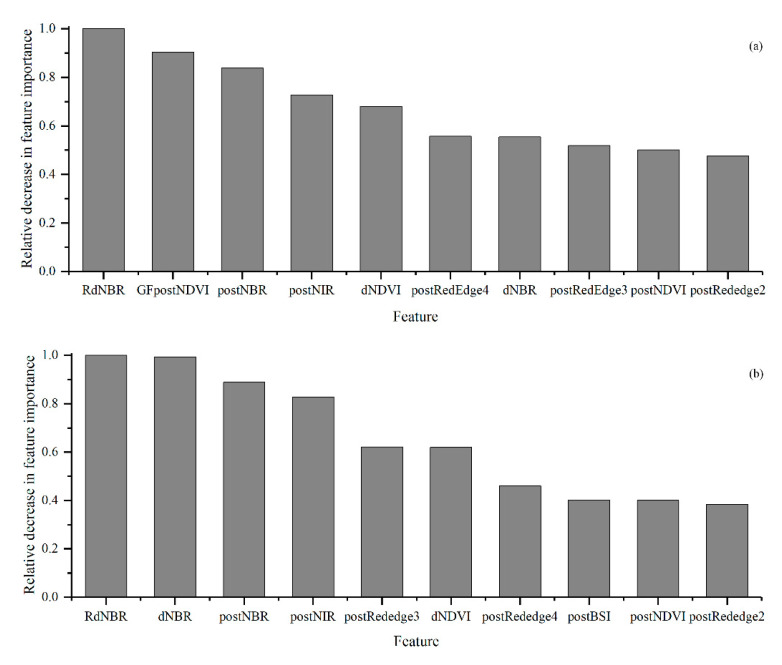
Ranked feature importance of each variable in the S2-GF model (**a**) and the S2 model (**b**), with 1 representing the most important feature. The figure only shows the top ten variables. Only GFpostNDVI, which means the NDVI of the post-fire GF image, was derived from GF images. The other features were obtained from Sentinel 2 images.

**Figure 6 sensors-23-02492-f006:**
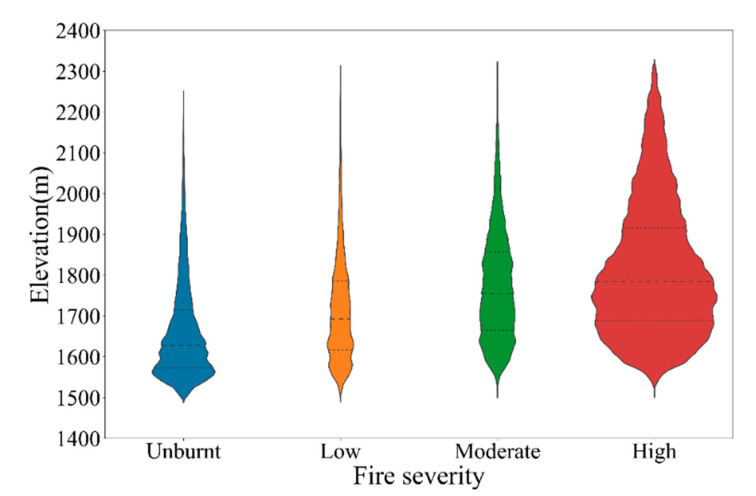
The distribution of elevation across each fire severity class. The violin density plots display the kernel density estimation of the underlying distribution, with the quartiles as horizontal lines (the median, the first, and the third quartile). Severity classes include unburnt, low severity (a burnt surface with unburnt canopy or partial canopy scorch), moderate severity (most or full canopy scorch), and high severity (full canopy consumption).

**Figure 7 sensors-23-02492-f007:**
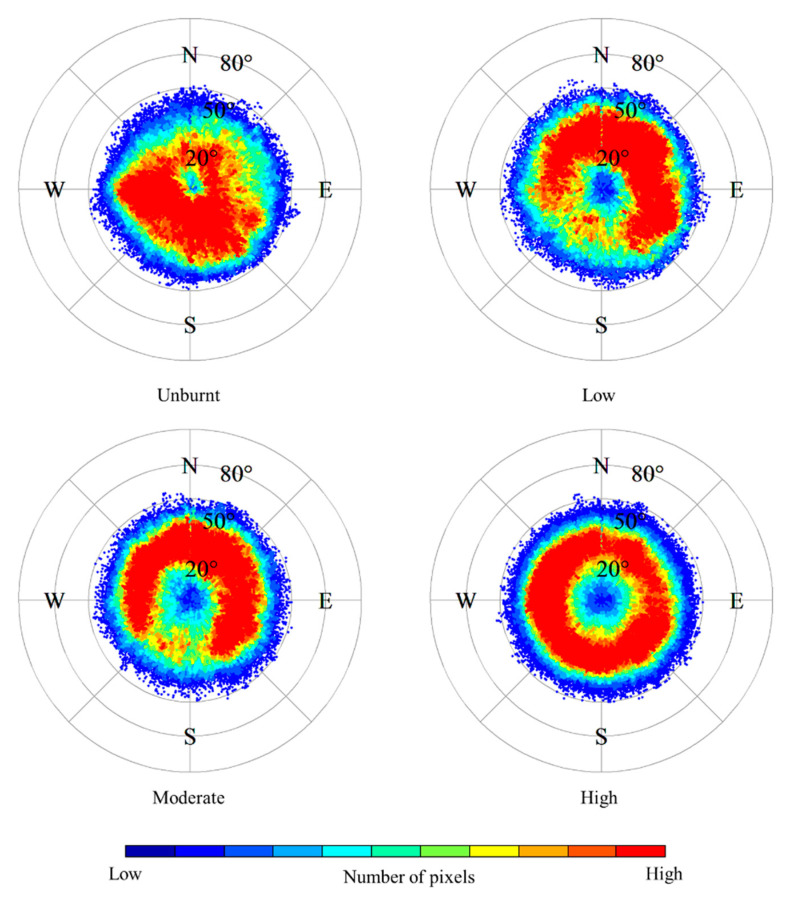
The distribution of aspect (denoted by angle) and slope (denoted by radius) across each fire severity class. The letters N, E, S, and W represent the directions of north, east, south, and west, respectively. Severity classes include unburnt, low severity (a burnt surface with unburnt canopy or partial canopy scorch), moderate severity (most or full canopy scorch), and high severity (full canopy consumption).

**Figure 8 sensors-23-02492-f008:**
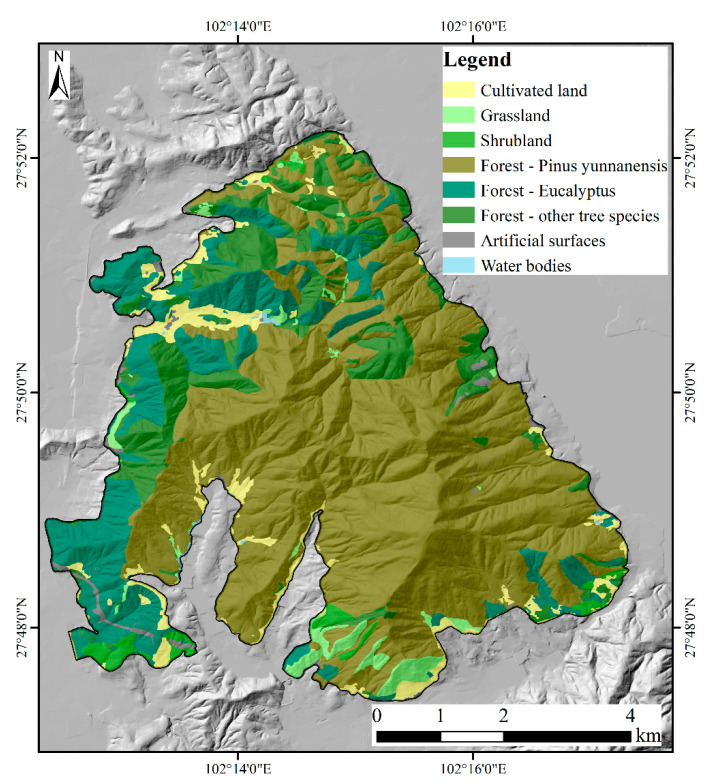
The distribution of land use types before the fire in the study area. The other tree species in the study area were mostly *Cyclobalanopsis*, aspen, and oak. The areas with higher altitudes are dominated by *Pinus Yunnanensis*, while the other vegetation was mainly distributed in the lower altitude areas in the west of the study area.

**Figure 9 sensors-23-02492-f009:**
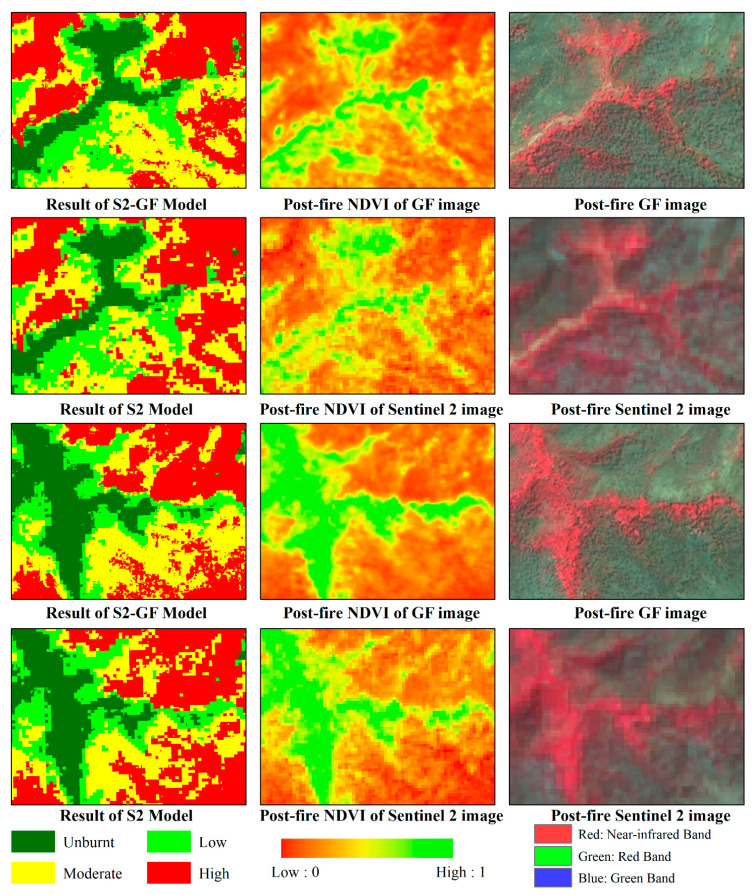
Comparison between the prediction results, post-NDVI features, and images of the two models. Due to the higher resolution, there were fewer mixed pixels in the GF image than in the Sentinel 2 image, especially in the transitional areas where the fire severity changes from low to high. The post-fire NDVI of GF images provided vegetation information with higher resolution, which can significantly improve the accuracy of fire severity classes in areas with highly variable fire severity classes.

**Table 1 sensors-23-02492-t001:** List of data used in this study.

Name	Pre-Fire Data Date	Post-Fire Data Date	Spatial Resolution
Sentinel 2 images	30 March 2020 (S2B)	4 April 2020 (S2A)	10 m/20 m
GF series images	6 March 2020 (GF-6)	18 April 2020 (GF-1)	2 m/8 m
Digital elevation Model	2013	2013	5 m
Plot-based vegetation maps	2019	2020	2 m

**Table 2 sensors-23-02492-t002:** Bands of GF-1 and GF-6.

Spectral Region	GF-1			GF-6		
	Band	Range (nm)	Spatial Resolution (m)	Band	Range (nm)	Spatial Resolution (m)
Pan ^1^	1	450–900	2	1	450–900	2
Blue	2	450–520	8	2	450–520	8
Green	3	520–590	8	3	520–600	8
Red	4	630–690	8	4	630–690	8
NIR ^2^	5	770–890	8	5	760–900	8

^1^ Pan: panchromatic. ^2^ NIR: near-infrared.

**Table 3 sensors-23-02492-t003:** Bands of Sentinel 2A and Sentinel 2B.

Spectral Region	Band	Sentinel 2A		Sentinel 2B		Spatial Resolution (m)
		Central Wavelength (nm)	Bandwidth (nm)	Central Wavelength (nm)	Bandwidth (nm)	
Blue	2	492.4	66	492.1	66	10
Green	3	559.8	36	559	36	10
Red	4	664.6	31	665	31	10
Red Edge 1	5	704.1	15	703.8	16	20
Red Edge 2	6	740.5	15	739.1	15	20
Red Edge 3	7	782.8	20	779.7	20	20
NIR ^1^	8	832.8	106	833	106	10
Red Edge 4	8A	864.7	21	864	22	20
SWIR ^2^ 1	11	1613.7	91	1610.4	94	20
SWIR 2	12	2202.4	175	2185.7	185	20

^1^ NIR: near-infrared. ^2^ SWIR: shortwave infrared.

**Table 4 sensors-23-02492-t004:** Classification of the fire severity, including the number of samples for each class.

Severity Class	Description	% Foliage Fire Affected	Number of Samples
Unburnt	Unburnt surface with green canopy	0% canopy and understory unburnt	138
Low	Burnt surface with unburnt canopy or partial canopy scorch	>50% green canopy <50% canopy scorched	63
Moderate	Most or full canopy scorch	>50% canopy scorched <50% canopy biomass consumed	82
High	Most or full canopy consumption	>50% canopy biomass consumed	217

**Table 5 sensors-23-02492-t005:** Features used in the models to identify fire severity.

Feature Type	S2-GF Model Trained with GF Data	S2 Model Trained without GF Data
Reflectance	The reflectance of each band of pre-fire and post-fire Sentinel 2 images	The reflectance of each band of pre-fire and post-fire Sentinel 2 images
	The reflectance of each band of pre-fire and post-fire GF images	
Spectral indexes	Spectral indexes of pre-fire and post-fire Sentinel 2 images	Spectral indexes of pre-fire and post-fire Sentinel 2 images
	Spectral indexes of pre-fire and post-fire GF images	
Additional data	DEM, aspect, slope, and vegetation type	DEM, aspect, slope, and vegetation type

**Table 6 sensors-23-02492-t006:** Spectral indexes calculated from Sentinel 2 images.

Spectral Index	Formula	References
Pre-Fire Normalized Burn Ratio (preNBR)	$\frac{\mathrm{preNIR}-\mathrm{preSWIR}2}{\mathrm{preNIR}+\mathrm{preSWIR}2}$	[18,19,20]
Post-Fire Normalized Burn Ratio (postNBR)	$\frac{\mathrm{postNIR}-\mathrm{postSWIR}2}{\mathrm{postNIR}+\mathrm{postSWIR}2}$	
The Differenced Normalized Burn Ratio (dNBR)	preNBR−postNBR	
Relative Differenced Normalized Burn Ratio (RdNBR)	$\frac{\mathrm{dNBR}}{\mathrm{sqrt}\left(\left|\mathrm{preNBR}\right|\right)}$	
Pre-Fire Normalized Difference Vegetation Index (preNDVI)	$\frac{\mathrm{preNIR}-\mathrm{preRed}}{\mathrm{preNIR}+\mathrm{preRed}}$	[35]
Post-Fire Normalized Difference Vegetation Index (postNDVI)	$\frac{\mathrm{postNIR}-\mathrm{postRed}}{\mathrm{postNIR}+\mathrm{postRed}}$	
The Differenced Normalized Difference Vegetation Index (dNDVI)	postNDVI−preNDVI	
Pre-Fire Bare Soil Index (preBSI)	$\frac{\left(\mathrm{preSWIR}1+\mathrm{preRed}\right)-\left(\mathrm{preNIR}+\mathrm{preBlue}\right)}{\left(\mathrm{preSWIR}1+\mathrm{preRed}\right)+\left(\mathrm{preNIR}+\mathrm{preBlue}\right)}$	[36]
Post-Fire Bare Soil Index (postBSI)	$\frac{\left(\mathrm{postSWIR}1+\mathrm{postRed}\right)-\left(\mathrm{postNIR}+\mathrm{postBlue}\right)}{\left(\mathrm{postSWIR}1+\mathrm{postRed}\right)+\left(\mathrm{postNIR}+\mathrm{postBlue}\right)}$	
The Differenced Bare Soil Index (dBSI)	postBSI−preBSI	

**Table 7 sensors-23-02492-t007:** Confusion matrix and accuracy statistic of the S2-GF model trained with the combination of S2 and GF series data and the S2 model trained without GF data.

		Reference from Field Investigation					F1 Score	Kappa
		Unburnt	Low	Moderate	High	Precision		
Predicted by the S2-GF model	Unburnt	26	2	0	0	92.86	94.55	0.8697
	Low	1	8	2	0	72.73	72.73	
	Moderate	0	1	15	1	88.24	83.34	
	High	0	0	2	42	95.45	96.55	
	recall	96.30	72.73	78.95	97.67	91.00	91	
Predicted by the S2 model	Unburnt	24	3	0	0	88.89	88.89	0.7808
	Low	3	6	2	0	54.55	54.55	
	Moderate	0	1	13	1	86.67	76.47	
	High	0	1	4	42	89.36	93.33	
	recall	88.89	54.55	68.42	97.67	85.00	85	

**Table 8 sensors-23-02492-t008:** Area statistics of fire severity class across each vegetation type.

Vegetation Type		Fire Severity				
		Unburnt	Low	Moderate	High	Total
The whole study area	Area(ha)	1017.62	430.63	597.74	2039.01	4085.00
	Percentage (%)	24.91	10.54	14.63	49.92	100.00
*Pinus Yunnanensis*	Area(ha)	356.67	339.38	497.35	1294.33	2487.73
	Percentage (%)	14.34	13.64	19.99	52.03	100.00
*Eucalyptus*	Area(ha)	195.74	50.13	63.34	341.96	651.17
	Percentage (%)	30.06	7.70	9.73	52.51	100.00
Other areas of the study area	Area(ha)	465.21	41.12	37.05	402.72	946.10
	Percentage (%)	49.17	4.35	3.91	42.57	100.00

## Data Availability

The data used in the study are available from the authors and can be shared upon reasonable requests.

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
