# Peer review of "Mapping Fire Severity in Southwest China Using the Combination of Sentinel 2 and GF Series Satellite Images"

_sensors, 2023, doi:10.3390/s23052492_

Round 1
Reviewer 1 Report
This work is related to fire severity estimation using multi-source high-resolution satellite images to improve global classification accuracy. However, in current mannuscript, the following questions are still unclear.
1. In a multi-band sallellite image, there are water, soil, rocks, forest plants and weeds, and other geo-objects. The current task just refers to vegetation classification, i.e., unburnt, moderate, and high severity classes, excluding those abiosis geo-objects, such as water, naked soils. It is unclear that how to remove those abiosis geo-objects, and how to evaluate the quality of removement.
2. Another question is about machine learning training. In supervised learning, training sample selection/construction and sample capacity are still unclear in the current manuscript. How to prove that the constructed sample patters are efficient for the classification. As far as classification accuracy are concerned, what is the ground-truth for the performance evaluation. The reviewer suggests sonme user-supplied pixel-level fire image annotation methods [1-2]. Forthemore, since the performance is evaluated on hetergeneous images, e.g., using the combination of Sentinel 2 and GF satellite images, for the discussed study area, how to realize image registration?
[1] Yang X, Chen R, Zhang F, et al. Pixel-level automatic annotation for forest fire image. Engineering Applications of Artificial Intelligence, 2021, 104: 104353.
[2] Wang H, Li Y, Dang L M, et al. Pixel-level tunnel crack segmentation using a weakly supervised annotation approach. Computers in Industry, 2021, 133: 103545.
Author Response
Response to Reviewer #1
This work is related to fire severity estimation using multi-source high-resolution satellite images to improve global classification accuracy. However, in current manuscript, the following questions are still unclear.
Response: Many thanks for your valuable suggestions of our manuscript. We sincerely appreciate the time you devoted to improving this paper. According to your suggestions below, we have revised the manuscript carefully.
- In a multi-band satellite image, there are water, soil, rocks, forest plants and weeds, and other geo-objects. The current task just refers to vegetation classification, i.e., unburnt, moderate, and high severity classes, excluding those abiosis geo-objects, such as water, naked soils. It is unclear that how to remove those abiosis geo-objects, and how to evaluate the quality of removement.
Response: Thank you for raising this important issue. To make this key issue clear, the land use interpretation results of the study area, with its accuracy and related discussions, were added in the revised manuscript.
Page 13, Line 279:
“The study area is a typical artificially planted forest ecosystem in Southwest China, as indicated by manual interpretation results of GF images with an accuracy of 98.68%. Forest covers 89.78% of the total land in the study area, while other land use types such as cultivated land, shrubland, grassland, artificial surfaces, and water bodies occupy 4.70%, 1.68%, 2.57%, 1.13%, and 0.14% respectively. Figure 8 shows the distribution of land use in the study area.”
Page 13, Figure 8:
“Figure 8. The distribution of land use types before the fire in the study area. The other tree species in the study area were mostly Cyclobalanopsis, aspen, and oak. The areas with higher altitudes are dominated by Pinus Yunnanensis, while the other vegetation was mainly distributed in the lower altitude areas in the west of the study area.”
Page 16, Line 363:
“The proportion of non-vegetative objects in the study area is small, accounting for just 1.27%. In areas where non-vegetative objects are more prevalent, it's recommended to include these objects, such as artificial surfaces, water bodies, and bare lands, in the training samples for the fire severity mapping model. This will reduce the potential impact of these objects on the accuracy of the predicted fire severity maps. It's important to consider both vegetative and non-vegetative components in the study area and make necessary adjustments to the model to ensure optimal performance.”
- Another question is about machine learning training. In supervised learning, training sample selection/construction and sample capacity are still unclear in the current manuscript. How to prove that the constructed sample patters are efficient for the classification. As far as classification accuracy are concerned, what is the ground-truth for the performance evaluation. The reviewer suggests some user-supplied pixel-level fire image annotation methods [1-2]. Furthermore, since the performance is evaluated on heterogeneous images, e.g., using the combination of Sentinel 2 and GF satellite images, for the discussed study area, how to realize image registration?
[1] Yang X, Chen R, Zhang F, et al. Pixel-level automatic annotation for forest fire image. Engineering Applications of Artificial Intelligence, 2021, 104: 104353.
[2] Wang H, Li Y, Dang L M, et al. Pixel-level tunnel crack segmentation using a weakly supervised annotation approach. Computers in Industry, 2021, 133: 103545.
Response: Thanks for your thoughtful review. We have taken all the papers into consideration in the revised manuscript. And these key issues you mentioned were further explained in the revised manuscript.
Page 14, Line356:
“In addition, the accuracy evaluation used a ground truth of the fire severity of 500 field-recorded samples. In regions with surveillance equipment, image annotation techniques can construct a large sample library from photos and videos to improve the accuracy of forest fire monitoring [39,40].”
Page 5, Line 157:
“The field survey recorded GPS coordinates, vegetation types, and forest canopy and understory coverage to classify the fire severity of sample points on-site, using the standard presented in Table 4.”
Page 7, Line 217:
“To prove that the constructed sample patterns are efficient for the random forest classification, evaluation metrics such as precision, recall, F1 score, confusion matrix, and Kappa statistic were used (Table 7).”
Page 8, Table7:
“Table 7. Confusion matrix and accuracy statistic of the S2-GF model trained with the combination of S2 and GF series data and the S2 model trained without GF data.”
|
|
|
Reference from field investigation |
F1 score |
Kappa |
||||
|
Unburnt |
Low |
Moderate |
High |
Precision |
||||
|
Predicted by S2-GF model |
Unburnt |
26 |
2 |
0 |
0 |
92.86 |
94.55 |
0.8697 |
|
Low |
1 |
8 |
2 |
0 |
72.73 |
72.73 |
||
|
Moderate |
0 |
1 |
15 |
1 |
88.24 |
83.34 |
||
|
High |
0 |
0 |
2 |
42 |
95.45 |
96.55 |
||
|
recall |
96.30 |
72.73 |
78.95 |
97.67 |
91.00 |
91 |
||
|
Predicted by S2 model |
Unburnt |
24 |
3 |
0 |
0 |
88.89 |
88.89 |
0.7808 |
|
Low |
3 |
6 |
2 |
0 |
54.55 |
54.55 |
||
|
Moderate |
0 |
1 |
13 |
1 |
86.67 |
76.47 |
||
|
High |
0 |
1 |
4 |
42 |
89.36 |
93.33 |
||
|
recall |
88.89 |
54.55 |
68.42 |
97.67 |
85.00 |
85 |
||
Page 5, Line 139:
“The high-resolution GF image was used as the reference for executing the image registration process. This ensured accurate alignment of the sentinel images to a consistent standard.”

Reviewer 2 Report
This study aims to map fire severity in Southwest China using satellite imagery. The combination of Sentinel 2 and GF series satellite images is used to gather data and analyze the fire severity in the region. This information can help in understanding the impact of fires on the environment and in making informed decisions for effective fire management.
1. Introduction: Well written providing a holistic overview of research, objective well defined.
2. Study Area: Well-defined, clear understanding of the study area depicted
3. Methodology: Well-structured, data, methods, and formulas used are well-written.
4. Results: Innovative, figures well presented, high resolution, the relationship between classes well depicted.
5. Discussion: Error analysis, feature importance comparison well represented.
6. Conclusion: Well written, depicting the major findings of the study
Minor comments for Improvement
1. Include a way forward in the conclusion as to how your study will be a role model for others missing, please include it.
rest ok
Author Response
Response to Reviewer #2
This study aims to map fire severity in Southwest China using satellite imagery. The combination of Sentinel 2 and GF series satellite images is used to gather data and analyze the fire severity in the region. This information can help in understanding the impact of fires on the environment and in making informed decisions for effective fire management.
- Introduction: Well written providing a holistic overview of research, objective well defined.
- Study Area: Well-defined, clear understanding of the study area depicted
- Methodology: Well-structured, data, methods, and formulas used are well-written.
- Results: Innovative, figures well presented, high resolution, the relationship between classes well depicted.
- Discussion: Error analysis, feature importance comparison well represented.
- Conclusion: Well written, depicting the major findings of the study
Minor comments for Improvement
- Include a way forward in the conclusion as to how your study will be a role model for others missing, please include it.
rest ok
Response: Many thanks for your positive evaluation of our manuscript. We sincerely appreciate the time you devoted to improving this paper. According to your valuable suggestion, we have revised the manuscript carefully.
Page 16, Line 379:
“In conclusion, this study presents an approach to map fire severity in Southwest China at 2 m spatial resolution using random forest models and satellite images from Sentinel 2 and GF series.”
Page 17, Line 393:
“The results of this study highlight the potential of high-resolution satellite images for mapping fire severity at fine spatial scales. This study presents a machine learning approach that combines the strengths of multiple satellite images, providing a reference for future research. Looking ahead, the future improvement of the satellite sensors, especially the high-resolution sensors with SWIR bands, can further improve the fire severity mapping accuracy. Continuing research to assess the sensitivity of different spatial scales of satellite images for mapping fire severity across various ecosystems is essential.”

Round 2
Reviewer 1 Report
Compared with the old version, this manuscript has been improved. The reviewer concerned questions have also been well answered.